# Automatic Analysis of Transverse Musculoskeletal Ultrasound Images Based on the Multi-Task Learning Model

**DOI:** 10.3390/e25040662

**Published:** 2023-04-14

**Authors:** Linxueying Zhou, Shangkun Liu, Weimin Zheng

**Affiliations:** College of Computer Science and Engineering, Shandong University of Science and Technology, Qingdao 266590, China

**Keywords:** artificial intelligence, computer aided analysis, deep learning, ultrasound, convolutional neural network, multi-task learning

## Abstract

Musculoskeletal ultrasound imaging is an important basis for the early screening and accurate treatment of muscle disorders. It allows the observation of muscle status to screen for underlying neuromuscular diseases including myasthenia gravis, myotonic dystrophy, and ankylosing muscular dystrophy. Due to the complexity of skeletal muscle ultrasound image noise, it is a tedious and time-consuming process to analyze. Therefore, we proposed a multi-task learning-based approach to automatically segment and initially diagnose transverse musculoskeletal ultrasound images. The method implements muscle cross-sectional area (CSA) segmentation and abnormal muscle classification by constructing a multi-task model based on multi-scale fusion and attention mechanisms (MMA-Net). The model exploits the correlation between tasks by sharing a part of the shallow network and adding connections to exchange information in the deep network. The multi-scale feature fusion module and attention mechanism were added to MMA-Net to increase the receptive field and enhance the feature extraction ability. Experiments were conducted using a total of 1827 medial gastrocnemius ultrasound images from multiple subjects. Ten percent of the samples were randomly selected for testing, 10% as the validation set, and the remaining 80% as the training set. The results show that the proposed network structure and the added modules are effective. Compared with advanced single-task models and existing analysis methods, our method has a better performance at classification and segmentation. The mean Dice coefficients and IoU of muscle cross-sectional area segmentation were 96.74% and 94.10%, respectively. The accuracy and recall of abnormal muscle classification were 95.60% and 94.96%. The proposed method achieves convenient and accurate analysis of transverse musculoskeletal ultrasound images, which can assist physicians in the diagnosis and treatment of muscle diseases from multiple perspectives.

## 1. Introduction

As an adjunct to clinical medicine, medical imaging is the basis for physicians to analyze pathological structures and plays a critical role in the diagnosis and treatment of disease. The types of medical imaging that are widely used in clinical practice include ultrasound imaging (ultrasound), Magnetic Resonance Imaging (MRI), Computed Tomography (CT), and X-ray imaging (X-ray). Ultrasound imaging is suitable for the structural assessment of most soft tissue organs in the body because it is painless, non-invasive, real-time, inexpensive, and free of ionizing radiation [1]. The results of imaging contain rich information on tissue structure and are a reliable indicator for the evaluation of pathological changes. Musculoskeletal ultrasound (MSUS) [2] is a new ultrasound technique for obtaining skeletal muscle imaging using high-frequency ultrasound to show the hierarchical relationships of soft tissues such as muscles and their internal structures. Skeletal muscle ultrasound images can demonstrate features such as muscle texture and muscle echogenicity. It can screen for potential neuromuscular diseases including myasthenia gravis, myotonic dystrophy, and ankylosing muscular dystrophy by looking at muscle status.

The quantitative analysis of longitudinal musculoskeletal ultrasound images enables us to obtain important parameters that reflect the state of the muscle, including pennation angle, muscle thickness, and fiber length. These morphological characteristics are directly related to the mechanical properties of muscle tissue and can guide rehabilitation science treatments such as muscle rehabilitation training and prosthetic control [3]. The analysis of transverse musculoskeletal ultrasound images enables the acquisition of muscle cross-sectional area (CSA) between deep and superficial aponeurosis and the early abnormal diagnosis of neuromuscular diseases. The correct segmentation of the muscle CSA is a key step in the assessment of the muscle condition. It reveals the strength level of muscles [4] and reflects the health level of sports and the severity of disease. In transverse musculoskeletal ultrasound images, normal muscles have moderate echogenicity, with reticular and banded separation visible in the middle of the deep tendon membrane, as well as speckled moderate to high echogenicity [5]. In contrast, pathologic muscles are affected by increased fat replacement and the presence of connective tissue, which can result in increased echogenic reflections that appear faintly textured, cloudy, or hairy and glassy [6]. This phenomenon has been found to correlate with the staging of neurological disease [7].

Since skeletal muscle ultrasound images contain complex noise, their analysis is a time-consuming and tedious process. The annotation of the muscle cross-sectional area requires the clinician to manually select the region of interest, and this manual operation is prone to errors [8]. In addition, pathological changes in the muscle result in a difficult distinction between the texture of muscle and the noise of ultrasound equipment, which adds to the difficulty of the physician’s diagnosis [9]. It requires a physician with extensive experience to perform early diagnosis accurately. Therefore, a computer-assisted method is needed to reduce the time and effort of clinicians in processing image information.

In this paper, we propose a method for transverse musculoskeletal ultrasound images based on multi-task learning. The method automatically segments muscle cross-sectional area from ultrasound images and provides a preliminary diagnosis of abnormal muscles. This automated analysis method can reduce the diagnostic burden of physicians and also provide an effective basis for subsequent related treatment. The method constructs a multi-task model based on multi-scale feature fusion and attention mechanism (MMA-Net). The network consists of a combination of U-Net and VGG, sharing a part of the layers in the shallow layer of the network and adding connection fusion information in the deep layer. The Atrous Spatial Pyramid Pooling (ASPP) module and coordinated attention (CA) module were added to the backbone network. The ASPP module can effectively increase the perceptual field of the shared network and segmentation branches for better integration of global features. The CA module adaptively calibrates attention for different axes of each branch, allowing the network to further focus on the features of each task. Compared with other single-task models, the method is more capable of extracting details and edge pixels and has better segmentation results. The classification results for abnormal muscles are more accurate and have good robustness.

The main contributions of this study are as follows.

We analyzed the existing methods for the analysis of longitudinal and transverse musculoskeletal ultrasound images, summarizing their advantages and limitations. We also discussed the clinical value provided by transverse musculoskeletal ultrasound images, as well as the difficulties of manual analysis.We proposed a multi-task learning-based analysis method for transverse musculoskeletal ultrasound images. The method achieves both the segmentation of muscle cross-section areas and the classification of abnormal muscles by training a multi-task learning model. For diseased and healthy muscle ultrasound images with complex noise, the addition of an attention module and a multi-scale fusion module effectively increase the accuracy of the results. Compared with the single-task learning approach, the proposed method can fully exploit the potential connection between two tasks and share additional information to enhance the ability of image analysis.We proposed a novel multi-task learning model, MMA-Net, which outperforms some single-task models on skeletal muscle ultrasound images and is stable and robust. In the future, it has the potential to be applied to the analysis of other ultrasound images of other organs.

The rest of this study is presented below. The second section describes the existing available methods for skeletal muscle ultrasound image analysis and summarizes their advantages and disadvantages. The proposed method and the structure of the MMA-Net are described in detail in Section 3. The fourth section describes the environment and parameter settings for conducting the experiments on the transverse musculoskeletal ultrasound image dataset. The fifth section presents and discusses the experimental results. The fifth section includes a summary of this work.

## 2. Literature Work

For the automatic analysis of skeletal muscle ultrasound images, many researchers have invested a lot of effort in the measurement of muscle structural parameters, and many classical methods have been proposed. Researchers first apply image processing techniques to transform the images, and then obtain quantitative or qualitative evaluation results using the proposed methods such as re-voting strategies, heuristic searches, and coordinate calculations. Zhou and Zheng [10] proposed a modified Hough transform method to identify the major muscle bundle directions in musculoskeletal ultrasound images, using a re-voting strategy to solve the blending problem in ultrasound image line detection with good results. Zhao et al. [11] investigated an automatic linear extraction method based on local Radon transform and a rotation strategy to detect parameters such as the angle of the muscle bundle in ultrasound images on this basis. This method achieves automatic measurement of the pennation angle, but it relies heavily on the selection of the edge detector in the image processing stage, the measurement of the image is often semi-automatic, and it is not applicable to ultrasound images with complex noise. Then, a method combining Gabor wavelet and Hough transform was proposed by Zhou et al. [12]. This is a method for the automatic identification of fibular muscle fin angle and muscle bundle length based on multi-resolution analysis and line feature extracting. This method performed well on simulated and real images with high scattered noise, but it relies on the setting of some parameters whose selection is empirical. As a fully-automated method for muscle thickness measurement, MUSA was proposed by Caresio and Salvi et al. [13]. The method detects the muscle bundles in the middle of the fascia using the Hough transform and performs an iterative heuristic search for the muscle bundles in the region of interest, from which the fascia contour is determined and the muscle thickness is calculated. Based on this approach, they proposed the first automatic algorithm for analyzing and segmenting muscle ultrasound images in the cross-sectional plane, TRAMA [14]. This method uses the Sobel operator and Gaussian filter to extract the fascia, then the image is further thresholded and a fast heuristic search is used to to reduce the number of aponeurosis candidates. Finally, the deep and superficial aponeurosis were filtered out and the cross-sectional area between them was calculated. These two methods achieve accurate measurements of important muscle parameters, but their processing is complicated, and the transformed or filtered images also need to set thresholds to filter noise, and they have weak generalization ability to ultrasound images with different echogenicity levels.

Recently, deep learning, a new research direction in the field of machine learning, has brought a new approach to learn the intrinsic laws and representation hierarchy of data [15]. Convolutional neural networks are one of the representative algorithms of deep learning. In the field of medical images, convolutional neural networks play an important role in disease prediction [16], organ segmentation [17,18], and lesion region identification [19] through effective learning of image information. Cunningham et al. [20] used convolutional neural networks for the first time to analyze muscle structure and proposed a model based on depth residuals and convolutional neural networks to measure the orientation and curvature of human muscle bundles. This was a novel attempt to use convolutional neural networks for ultrasound image analysis of skeletal muscle. After this, this method was further improved [21]. The deconvolution and maximum deconvolution DCNNs were used to quantify muscle parameters, achieving relatively robust parameter estimation. However, the error of the pennation angle of this method is as high as 6°, and there are still some deficiencies in the measurement accuracy. Kompella et al. [22] used R-CNN for segmentation of knee cartilage, using 256 images acquired from only one volunteer on two different angles of the right knee sequence. On 55 test images, the final DSC was 80%. After this, an automatic method for measuring the pennation angle based on convolutional neural networks and active contours was proposed by Zheng et al. [23]. This method first uses a local Radon transform to detect the fascicle, then introduces a reference line to help detect the direction of the muscle bundle. Finally, the pennation angle is calculated. In 2022, Zheng et al. [24] proposed a fully automated method for muscle parameter analysis, which is based on the accurate segmentation of ultrasound images by the depth residual contraction U-Net (RS-Unet). Then, processing and calculation are performed to obtain three muscle parameters. Compared with existing methods for longitudinal ultrasound image analysis of skeletal muscle, the effect of complex noise on image segmentation accuracy is effectively eliminated, and the measurement of muscle parameters is very comprehensive and accurate.

Regarding the specific task of skeletal muscle ultrasound image analysis, most analyses have focused on the measurement of parameters in longitudinal muscle ultrasound images. In recent years, several analysis methods based on transverse muscle ultrasound images and diagnostic discrimination of inflammatory abnormal muscle pathology by convolutional neural networks have emerged. Burlina et al. [8] explored a method to automatically diagnose myositis using deep convolutional neural networks (DL-DCNNs) with an accuracy of about 79.2% in the classification of myositis in different regions of muscles. This method demonstrated the excellent performance of convolutional neural networks in automatic medical image classification and laid the foundation for subsequent studies on the classification of abnormal muscles. Marzola et al. [1] developed a method to segment the cross-sectional area (CSA) of transverse skeletal muscle ultrasound images using a convolutional neural network (CNN). The CNN was used to segment the image and post-processing of the output was used to obtain a finer segmentation. Since the cross-sectional area boundaries of abnormal muscles are more blurred and their segmentation is more difficult, the accuracy of the segmentation results reached 93% on normal muscle height but only 80% on abnormal muscles. Based on this, Marzola et al. [25] further investigated a method to diagnose abnormal muscles based on the gray level of the cross-sectional area. The method firstly segments the cross-sectional area (CSA) using a combination of multiple convolutional neural networks. Next, the average gray level z-score of the segmented portion is calculated, and the z-score that evaluates the muscle health level is used to determine whether the muscle is diseased or abnormal. This research method has made some progress compared with previous studies, and the accuracy of the classification result of abnormal muscle is about 91.5%, but the segmentation result of muscle cross-sectional area is still only 90%, which needs to be optimized.

With the development of deep learning, the emergence of multi-task learning has enabled sufficient information sharing among related tasks to enhance the learning efficiency of the network [26]. In the field of medical image processing, due to the rich information contained in images, multiple analysis tasks are intrinsically connected with each other. Single-task learning cannot tap the relationship between tasks and obtain additional useful information. Moreover, when facing more complex problems, the task can only be decomposed into multiple subtasks for training, which is tedious and wasteful of resources. Therefore, researchers have tried to apply a multi-task learning framework to solve clinical medical problems. These network models can implement multiple image processing tasks simultaneously and provide new ideas for the intelligent analysis of medical images. Zhao et al. [27] proposed a multi-task collaborative model, MCL-Net, for multi-metric quantification of the optic nerve head. The method is capable of simultaneously segmenting and classifying the optic nerve in fundus images. The representations of the two branches are exchanged and aggregated between the two tasks via a functional interaction module, FIM, for mutual collaboration. Chen et al. [28] proposed a multi-task U-Net model for skin melanoma detection, which improved on the U-Net model by adding two branching structures for classification at the bottom of the U-shaped structure and in front of the output layer. This classification structure can help determine whether melanoma is present in the skin, and thus decide whether segmentation is needed. Segmentation can assist in determining the lesion area and improve the accuracy of the classification results. Hugo Michard et al. [29] estimated muscle bundle angles and bundle lengths from ultrasound images using a new vector field model of bundle structure and a new multi-tasking neural network architecture, AW-Net. This approach uses a modified U-Net with attention gates to accurately estimate muscle structure and properties in a fully automated manner. There have been attempts by researchers to apply multi-task learning methods to analyze skeletal muscle ultrasound images. However, existing multi-task learning methods do not involve the analysis of transverse ultrasound images of skeletal muscle.

In summary, the existing analysis methods based on image processing and single-task learning suffer from insufficient accuracy and single analysis parameters. Therefore, in order to overcome the shortcomings of existing methods and explore more advanced analysis methods applicable to skeletal muscle ultrasound images, we proposed a novel multi-task learning method to solve the problem of automatic analysis of transverse ultrasound images of skeletal muscle. The proposed method concisely obtains accurate muscle pathology information which can assist physicians in further analysis and diagnosis.

## 3. Methods

In this paper, our proposed method for the analysis of transverse ultrasound images of skeletal muscle was implemented by a multi-task learning algorithm. The framework diagram is shown in Figure 1. First, the dataset containing segmentation labels and classification labels was fed into the MMA-Net for training. Second, the validation results were used during the training process to adjust the network hyperparameters and construct the optimal model. Finally, a trained neural network model was used to obtain segmentation maps of muscle cross-sections and classification results of abnormal muscles. This analysis method based on multi-task learning can obtain both pathological information for a comprehensive analysis of skeletal muscle transverse ultrasound images. The specific structure of the MMA-Net in the proposed analysis method is described below.

### 3.1. Network Architecture

MMA-Net is a multi-task learning based network model for the analysis of two pathological information in transverse musculoskeletal ultrasound images. The model extracts shared features in the shallow layer and constructs two branches in the deep layer to learn task-specific features separately. Information transfer is performed between the deep layers to supplement feature information. Figure 2 illustrates the specific structure of the proposed multi-task learning model.

The backbone structure of the model is composed of a combination of U-Net [30] and VGG [31] network models. The network takes a 512 × 512 size image as input and uses the structure of U-Net in the encoder part of the network. Meanwhile, jump connections are made between shrinking and expanding paths to propagate contextual information to higher resolution layers. The shallow network of the model fully shares parameters to extract global features for segmentation and classification. After the encoder goes through three layers of feature extraction, a classification branch is added to extract high-level semantic features for the classification task. The classification branch was designed to be similar to the VGG network, with alternating convolutional and pooling layers to continuously extract features and reduce image size, and finally outputs two neurons through the fully connected layer, representing muscle normal and abnormal, respectively. After continuing the downsampling in two layers, the split branch uses the same decoder as U-Net to recover the image details and complete the upsampling of the image. The final output is a feature map of size 512 × 512, which is used as the segmentation result after the sigmoid [32] function.

Between the segmentation and classification branches, feature fusion is performed by replication and splicing operations to supplement the spatial information for the classification task. This feature fusion can help the classification task to better abstract the semantic information of the region and effectively improve the accuracy of the classification task. Finally, a residual block [33] was used to replace the normal convolutional block in the backbone network. This improvement enhances the feature extraction capability of the network and avoids the degradation problem caused by the network being too deep.

### 3.2. The Atrous Spatial Pyramid Pooling (ASPP) Module

In order to increase the network’s ability to obtain global information and maximize the receptive field, after the residual blocks of the second, third and fourth levels of the encoder, an atrous spatial pyramid pooling (ASPP) module [34] was added. The structure of the module is shown in Figure 3.

The multi-scale feature information was extracted by combining the feature maps of different receptive fields using atrous convolution and pooling in parallel with atrous rates of 1, 6, 12, and 18, respectively. For the different sizes and locations of the cross-sectional region of skeletal muscle transverse ultrasound image and the complex texture information in this region, the atrous convolution in the ASPP module expands the receptive field and the multiscale fusion enhances the breadth of feature extraction. In the shared layer part, such a module serves to enhance the global information acquisition and ensures the extraction of low-level shared features for both tasks. In the segmentation branch, the fusion of different scale feature maps plays an important role in the integrity of the segmented region due to the large noise in the image and unclear segmentation edges.

### 3.3. The Coordinate Attention (CA) Module

After the global feature extraction in the shared layers, we added a coordinated attention (CA) module [35] in the segmentation and classification branches. This module is used to extract key features for each branch for a specific task. The structure of the CA module is shown in Figure 4.

It has a similar structure to the Squeeze and Excitation (SE) Attention module, which is divided into two parts, squeezing and attention generation. They are used for coordinate information embedding and adaptive recalibration of coordinate attention. The squeezing part stimulates the attention block to capture long-range interactions spatially using positional information by the average pooling over the X and Y coordinate axes, respectively. Specifically, given input *X*, the output of the *c*-th channel at height *h* and width *w*, obtained by encoding each channel along the horizontal and vertical coordinates using two spatially scoped pooling kernels (H,1) and (1,W), can be expressed as [35]:(1)$$z_{c}^{h}\left(h\right)=\frac{1}{W}\sum_{0\leq i\leq W}x_{c}\left(h,i\right)$$
(2)$$z_{c}^{w}\left(w\right)=\frac{1}{H}\sum_{0\leq j\leq H}x_{c}\left(j,w\right).$$

These two transformations allow the attention block to capture long-range dependencies along one spatial direction and retain precise location information along the other spatial location direction, helping the network to locate objects of interest more accurately. The pooling tensor is stitched for both directions and shrunk by a nonlinear transformation. Finally, it is split into two feature maps according to the spatial dimension, and the weights are learned through the convolutional layer. The output is extended by a sigmoid function and used as attention weights. The addition of CA attention module not only makes the network focus on the attention of the channel domain, but also considers the encoding of spatial information. It can effectively focus attention on the abstraction of texture features in the classification branch and enable the network to focus more on the segmentation of target regions and the extraction of edge space information in the segmentation branch. This module helps the network to better extract features specific to segmentation and classification after shared feature learning, while improving the network’s prediction accuracy for two tasks.

### 3.4. Combined Loss Function

The MMA-Net needs to learn both unique and shared features of two tasks, so we proposed a combined loss function. The weighted sum of the loss functions of the two tasks was used as the combined loss function, which is formulated as follows:(3)$$Loss_{multi}=\lambda L_{seg}+L_{cla},$$
where $L_{seg}$ is the loss function for the segmentation task and $L_{cla}$ is the loss function for the classification task. λ is the balance factor in the combination function. Since the complexity of the two tasks is different and there are differences in the rate of gradient descent and the order of magnitude between the loss functions, the weight factor λ was introduced to balance the weights and orders of magnitude of the loss functions for the different tasks. Adjusting the weight factor prevents the situation in which the network is significantly more biased for one task than the other, allowing both tasks to be trained better. In the combined loss function, Diceloss [36], which is commonly used for medical image segmentation, was used as the loss function for the segmentation task. The formula is as follows:(4)$$L_{seg}=Diceloss=1-\frac{2\sum_{i=1}^{N}y_{ij}{\hat{y}}_{ij}}{\sum_{i=1}^{N}y_{ij}+\sum_{i=1}^{N}{\hat{y}}_{ij}},$$
where $y_{ij}$ is the true probability value for each pixel, ${\hat{y}}_{ij}$ represents the predicted probability value, *N* is the total number of pixels.The loss for the classification task used a cross-entropy function [37], which is calculated as follows:(5)$$L_{cla}=CEloss=-\frac{1}{N}\sum_{i=1}^{N}t_{i}\log_{}p_{i}+\left(1-t_{i}\right)log_{}\left(1-p_{i}\right),$$
where *t* and *p* represent the true category and the corresponding predicted output, respectively.

The hyperparameter λ in the combined loss function was continuously adjusted during the model training process to observe the changes in the performance metrics of the segmentation and classification task. It was finally found that both tasks were well trained when λ = 20 and the performance of the proposed model was optimal.

## 4. Experiments

### 4.1. Datasets

The samples used in experiments were from the public datasets of transverse musculoskeletal ultrasound images in the paper by Marzola et al. [25]. The datasets were acquired during routine clinical practice, at the Radboud University Medical center. Images were scanned transversely by experienced muscle neurodiagnostic technicians using ultrasound equipment on muscle tissue. Data were analyzed from multiple subjects with images of different parts of the muscle and, for each subject, three or four transverse ultrasound images of skeletal muscle were acquired as required by the clinical protocol. The medial gastrocnemius muscle ultrasound images, which are of more clinically generalizable value in this dataset, were selected for our experiments. Of these samples, 349 images were from diseased subjects and another 1478 were from healthy subjects, for a total of 1827 images. Healthy muscles show moderate echogenicity in transverse ultrasound images of muscle, with reticular and band-like separations and speckled echogenicity in the middle of deep and superficial tendon membranes, which are myofascicular and intramuscular structures. In contrast, the diseased muscles showed blurred texture and echogenic enhancement between the deep and superficial tendon membranes, with a cloudy or hairy glass-like morphology. Figure 5 shows the transverse muscle ultrasound images of healthy and diseased subjects, respectively.

Labels of the datasets consisted of two parts, one part was the muscle classification label for whether the muscle is abnormal or not, and the other part was the annotated muscle cross-sectional segmentation map. Since the segmentation of the muscle cross-sectional area in the paper by Marzola et al. [25] is mainly used to calculate the average gray value of the region, to eliminate the influence of edge regions on the grayscale calculation, they cropped and shrank the annotated regions, as in Figure 6b. This annotation does not accurately represent the muscle cross-sectional area. Therefore, based on the definition of muscle cross-sectional area, edge expansion and regional extension in the aponeuroses direction were performed with FIJI software [38] based on the original labeling. The labeling results are shown in Figure 6c. The annotated data after regional expansion have been reviewed by a medical professional. The data containing the classified and re-labeled segmentation labels were used as the dataset for the training and testing of the model.

### 4.2. Implementation Details

All experiments were conducted on the server. The hardware platform and operating environment of the experiment are shown in Table 1. During the experiments, a total of 1827 images were used, of which 10% were randomly selected as the test set, 10% as the validation set, and the remaining 80% as the training set. The input image size of the model was 512 × 512 pixels and the batch size was set to 2. As an optimization strategy, we used the SGD [39] optimizer with decay and a momentum of 0.9. The initial learning rate was 1 × 10${}^{-4}$. The learning rate was adaptively adjusted using the ReduceLROnPlateau function, which reduces the learning rate according to the change in accuracy, with patience set to 10. The model was trained for a total of 50 epochs. When the Dice coefficients and accuracy rates tended to be stable, the model terminated the training.

### 4.3. Evaluation Metrics

In order to objectively reflect the performance of the method, we evaluated the proposed model using various evaluation metrics. The evaluation metrics for segmentation results are Dice Similarity Coefficient (DSC), Intersection over Union (IoU) and Pixel Accuracy (PA) [40]. From the perspective of calculating the region similarity, DSC and IoU were used to evaluate the distance difference between the segmentation result and the ground truth. The following are their formulas.
(6)$$DSC=\frac{2|X\cap Y|}{\left|X|+|Y\right|}$$
(7)$$IoU=\frac{\left|X\cap Y\right|}{\left|X|+|Y|-|X\cap Y\right|},$$
where *X* is the ground truth and *Y* is the area output predicted by the model. The comparison of multiple metrics can evaluate the segmentation performance of the model in many aspects and prove the effectiveness of the model to the maximum extent. PA is the percentage of correctly classified pixels in the image, which allows a more detailed evaluation of the segmentation results. Its calculation is as follows:(8)$$PA=\frac{\sum_{i=0}^{n}p_{ij}}{\sum_{i=0}^{n}\sum_{j=0}^{n}p_{ij}},$$
where *n* represents the total number of categories, $p_{ii}$ and $p_{ij}$ are the total number of pixels whose real pixel category is *i*, which are predicted as category *i*, and the total number of pixels whose real pixel category is *i*, which are predicted as *j*.

The Accuracy, Precision, Recall, and F-Score were used to evaluate the classification results [41]. The calculation formulas of the four evaluation indicators are as follows:(9)$$Accuracy=\frac{TP+TN}{TP+TN+FP+FN}$$
(10)$$Precision=\frac{TP}{TP+FP}$$
(11)$$Recall=\frac{TP}{TP+FN}$$
(12)$$F_{score}=\frac{2\times Precision\times Recall}{Precision+Recall},$$
where TP represents the number of negative samples classified as positive samples, FN represents the number of classification errors in negative samples, TN represents the number of correct classification in positive samples, and FP represents the number of classification errors in negative samples. The experiment also introduced AUC to comprehensively evaluate the classification performance of the model. The AUC is obtained by summing the area under the ROC curve. The ROC curve mainly focuses on two indicators: true positive rate (TPR) and false positive rate (FPR) [42]. These two indicators are the vertical and horizontal coordinates of the curve, the formulas are:(13)$$TPR=\frac{TP}{TP+FN}$$
(14)$$FPR=\frac{FP}{FP+TN}.$$

## 5. Results and Discussion

### 5.1. Ablation Study

To investigate the effectiveness of multi-task learning in skeletal muscle ultrasound image analysis, experiments were conducted to compare the performance differences between single-task and multi-task backbone models under the same dataset conditions. Table 2 shows the segmentation and classification performance. In the experiments of the single-task model, the network structure used was identical to the structure of one of the branches (classification or segmentation branch) of the multi-task backbone model, and the parameters set for the experiments were the same. It is observed in Table 2 that, in terms of segmentation, the DSC and IoU of the multi-task backbone model increased by 4.39% and 6.53%, respectively, compared with the single split task. In terms of classification, the accuracy of the multi-task was improved by 6.59% over the single-task classification. This demonstrates that the classification and segmentation tasks of transverse ultrasound images of skeletal muscle play a positive role in training each other. Multi-task learning can take advantage of the intrinsic connection that exists between the two tasks and enhances the network’s ability to mine potential features, which has the effect of optimizing the segmentation and classification performance of the model.

On the multi-tasking backbone, we also added modules to improve the network performance. The ablation experiments were carried out on each module of the model, and the effects of multi-scale feature fusion and attention module on the proposed model were analyzed to demonstrate that each module is important for both segmentation and classification results of MMA-Net. In the ablation experiments, we showed the results of each model on the test set. Table 3 and Table 4 show the impact of each module and connection structure for the segmentation and classification tasks.

As can be seen from the tables, the addition of the ASPP module enlarges the receptive field of the network compared to the backbone and enhances the global feature extraction ability at different scales. As a result, the DSC and IoU are improved by 3.27% and 5.36%, respectively, in the segmentation task, and the accuracy is improved by 2.19% in the classification task. The addition of the CA module to the network improved both network segmentation and classification performance, with a 2.94% improvement in F-score for the classification task. This is due to the ability of the CA module to focus attention on a specific task in each branch, enhancing the ability of the task to learn key features. The connection structure, as a soft parameter sharing mechanism, can transfer the spatial information of the segmentation network to the classification network, resulting in an enhanced performance of the classification task. Based on the characteristics of skeletal muscle ultrasound images, we finally constructed the structure of MMA-Net. It has a segmentation Pixel Accuracy (PA) of 97.91%, and a classification F-score and AUC of 93.95% and 97.62%, respectively, on the test set. The experiments demonstrate that adding multi-scale modules, attention modules, and connection structures in branches to the backbone can effectively improve the segmentation and classification capabilities of the network. The proposed model can well analyze skeletal muscle transverse ultrasound images from multiple angles, providing accurate regional information of CSA and classification results of abnormal muscles.

To analyze the generalization ability and the risk of overfitting of the MMA-Net, we conducted a 10-fold cross validation on the dataset. Table 5 shows the mean and standard deviation of the cross validation on each validation set. As seen in the table, the evaluation scores of the MMA-Net for both segmentation and classification are similar to the results on the test set. The standard deviations of the scores of the 10 results are also small, including 1.17% for DSC and 1.75% for Accuracy. The experiments demonstrate that the MMA-Net has good generalization ability.

### 5.2. Shared Layer Study

Due to the correlation between two tasks, the proposed model applies a more concise hard parameter sharing mechanism in the shallow network, which completely shares a part of the network layer. The advantage of hard parameter sharing is that it can widely obtain the common features between tasks, reduce the risk of overfitting the model, and is less computationally intensive. However, there are certain requirements for the number of shared layers, and the choice of the number of hard parameter sharing layers varies according to different tasks. The appropriate number of shared layers maximizes the acquisition of global features associated with multiple tasks and preserves the learning of features unique to each task. To investigate the optimal number of shared layers, shared layer experiments were conducted. As with U-Net, one pooling and one residual block were considered as a layer, and we attempted to adjust the shared layers from 0 to 5, respectively, to observe the performance changes of segmentation and classification.

Figure 7a shows the effect of the number of shared layers on the segmentation results. The change of the three metrics shows that the segmentation performance tends to increase with the number of shared layers. The best segmentation performance of the model is achieved when the number of shared layers is three, and decreases slightly thereafter. Figure 7b shows the results of the classification task. The classification effect is poor when the shared layer is zero, and the classification performance is best when the number of shared layers is three. The accuracy and other indicators decrease after the shared layers continue to increase, and the recall rate remains stable. This indirectly indicates that there is some correlation and difference between the two tasks. Sharing a portion of network layers will improve the results of both tasks, but too many shared layers will affect the extraction of unique features for each task. Therefore, we set the number of shared layers of the model to three. This allows MMA-Net to fully utilize the shared features of both tasks while balancing the learning of unique features for each task.

### 5.3. Comparison with Other Models

To evaluate the proposed methodology, the MMA-Net was compared with several classical segmentation and classification models for experiments. Table 6 shows the results of proposed model and other single-task models on the skeletal muscle ultrasound image datasets. As can be seen from the table, the proposed model achieves both the segmentation and the classification of skeletal muscle ultrasound images, reducing the cost of training multiple models. Compared with several classical single-task models, the results of both segmentation and classification were more accurate. In terms of classification, MMA-Net improves the performance a lot over several single classification models. The F-score and AUC were also improved by 6.04% and 3.91% compared to GoogleNet, the best classification model among the comparison models. This is due to the fact that muscle abnormalities are more difficult to distinguish in the representation of ultrasound images, and usually single classification networks have limited ability to extract abnormal features. However, multi-task learning allows the classification task to tap more potential features. In addition, the multi-scale feature fusion and attention mechanisms enable the model to have enhanced extraction of global and key features, resulting in a significant improvement in the classification ability for abnormal muscles. In terms of segmentation, the IoU and PA of MMA-Net are improved by 1.67% and 2.46%, respectively, compared to U-Net. It also improves compared to other advanced segmentation models. Because most of the multi-tasking models are only applicable to solving specific problems, and the network structure and functions of each model are very different, we did not compare the proposed network with other multi-tasking networks.

In order to observe the effect of the proposed model in the segmentation task more clearly, we visualized the segmentation results. The segmentation results of skeletal muscle images of healthy and diseased subjects by U-Net, U-Net++, LinkNet, DeeplabV3+ and the proposed model are shown in Figure 8 and Figure 9. As can be seen in Figure 8, the healthy skeletal muscle images have a clearer and easier structure for segmentation. The proposed model is more accurate than the other models in segmenting the details and handles the edges better. For diseased skeletal muscle ultrasound images, the CSA is more difficult to distinguish from other parts with similar contrast. In Figure 9, the segmentation results of some diseased images by other models are quite different from the ground truth. In contrast, our model can also effectively extract key features in noisier and more complex images of diseased skeletal muscle, achieving segmentation with high completeness and accuracy. This is due to the multi-task learning mechanism in MMA-Net, which supplements the model with association information between two tasks. Additionally, the multi-scale feature fusion and attention mechanisms make the models much more capable of feature extraction. The proposed model was experimentally shown to be robust with better segmentation results on both healthy and diseased muscles.

To further analyze the statistical significance of the results, *t*-tests were performed between MMA-Net and the other models. The *t*-test is used to analyze whether the difference between the means of two samples and the totalities they each represent is significant. When the *p* < 0.05, it indicates that there is a significant difference between the two totalities. We performed four random divisions of the dataset to form four datasets for training and testing each model, respectively. We first performed the Shapiro–Wilk test on each group of test results. The results show that the *p*-values for each group of data are greater than 0.05 and the assumption of normality is accepted. This indicates that the data follow a normal distribution and a *t*-test can be performed. Table 7 and Table 8 show the *t*-test results of the Dice coefficients and the accuracy of MMA-Net compared to the other models on the four test sets. As shown in the table, the *p*-values between MMA-Net and the other models are less than 0.05. This indicates that the performance improvements of MMA-Net in both segmentation and classification are statistically significant.

### 5.4. Comparison with Existing Methods

To further validate the proposed method, we compared it with existing methods for cross-sectional area segmentation and abnormality classification on transverse ultrasound images of skeletal muscle. The method we compared was proposed by Marzola et al. [25] in 2021. It first segmented the transverse ultrasound image of skeletal muscle using a convolutional neural network to determine the muscle cross-sectional area (CSA), and then further calculated the average gray value of this area as a criterion for determining abnormal muscle. Table 9 shows the evaluation results of the two methods for muscle cross-sectional area segmentation and abnormal muscle classification. As can be seen from the table, our method shows a large improvement in both segmentation and classification compared to the method of Marzola et al. [25]. The Intersection over Union between our method and the real label is 0.94, and the recall of abnormal image is also as high as 0.95. Figure 10 shows the indicators of the segmentation and classification results clearly in the form of bar graphs. In general, compared with the existing analysis methods, the proposed method based on MMA-Net is not only more concise, but also greatly improves the accuracy of the analysis results. It can achieve more accurate muscle cross section segmentation and abnormal muscle detection.

## 6. Conclusions

In this paper, a method based on MMA-Net for the analysis of transverse musculoskeletal ultrasound images was proposed to solve the problem of abnormal classification and cross-sectional area acquisition of transverse muscle ultrasound image. The approach was implemented by a multi-task model and reduces the cost of training multiple neural networks. The proposed model exploits the correlation between two tasks to mine potential features and uses parameter sharing mechanisms to enhance the generalization ability of the network. The multi-scale feature fusion and attention modules incorporated in the model enhance the feature extraction capability of the network and expand the perceptual field of the shared network layer. The proposed model exploits the correlation between two tasks to mine potential features and uses parameter sharing mechanisms to enhance the generalization ability of the network. We discussed the effectiveness of the MMA-Net network structure, evaluated with experiments. Additionally, comparisons with other single-task models and existing methods were made. Experimental results demonstrate that the proposed model is more capable of extracting edge detail features in terms of segmentation. It also has good robustness for diseased skeletal muscle images with blurred tissues. In terms of classification, the proposed model can better learn key texture features with higher accuracy and recall. In summary, the proposed method is robust and can achieve accurate skeletal muscle cross-sectional area segmentation and abnormal muscle classification. In the future, we will try to apply the proposed model to the analysis of other organ tissues. We are also exploring its application in clinical medicine systems to assist physicians in obtaining accurate and effective pathology information more concisely.

## Figures and Tables

**Figure 1 entropy-25-00662-f001:**
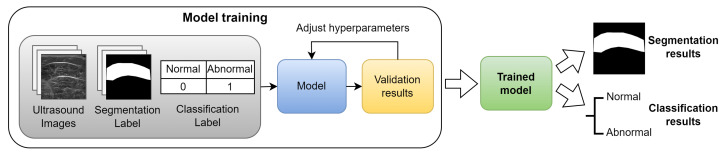
Framework diagram of the proposed method.

**Figure 2 entropy-25-00662-f002:**
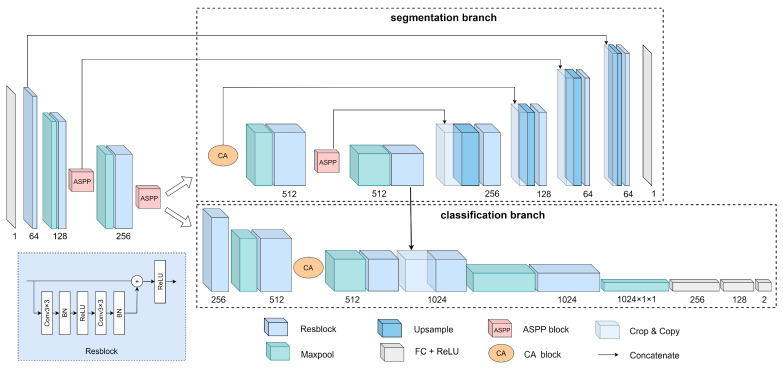
The structure of MMA-Net.

**Figure 3 entropy-25-00662-f003:**
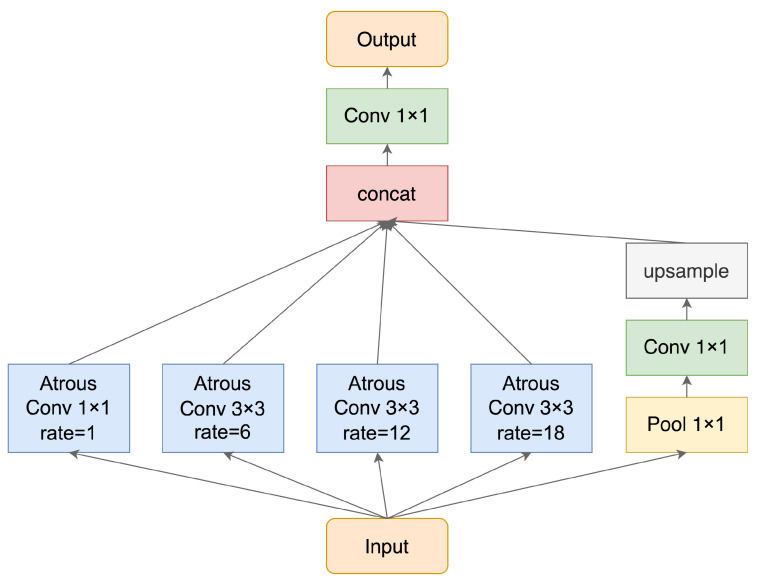
The structure of ASPP module.

**Figure 4 entropy-25-00662-f004:**
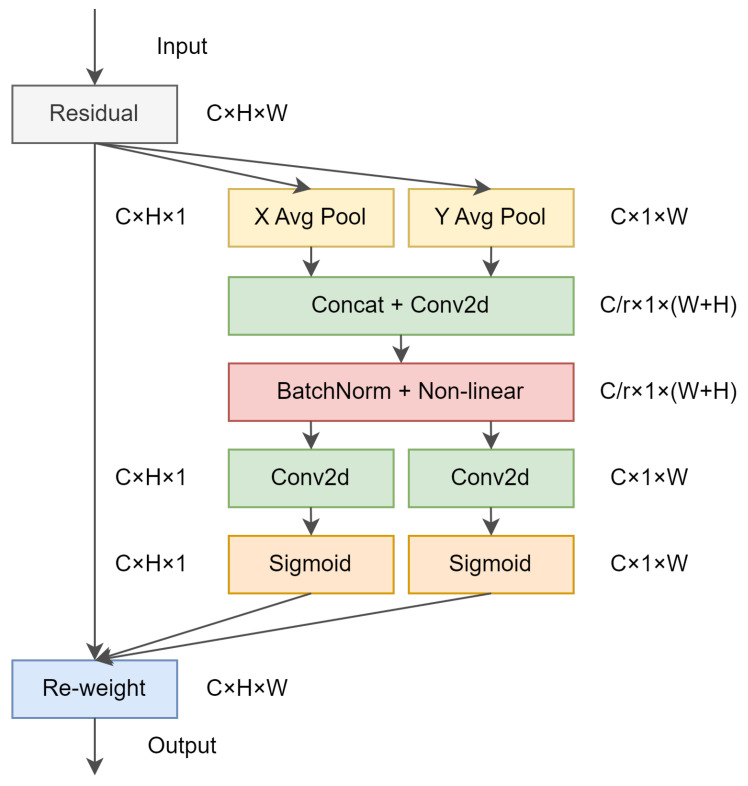
The structure of the CA module.

**Figure 5 entropy-25-00662-f005:**
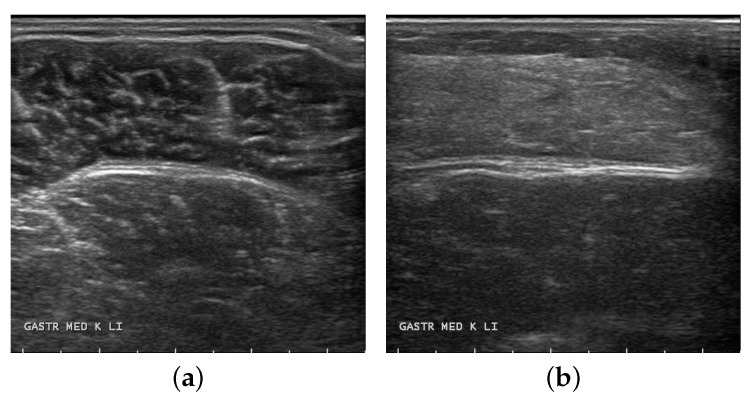
Samples of transverse ultrasound muscle images in healthy (**a**) and diseased (**b**) subjects.

**Figure 6 entropy-25-00662-f006:**
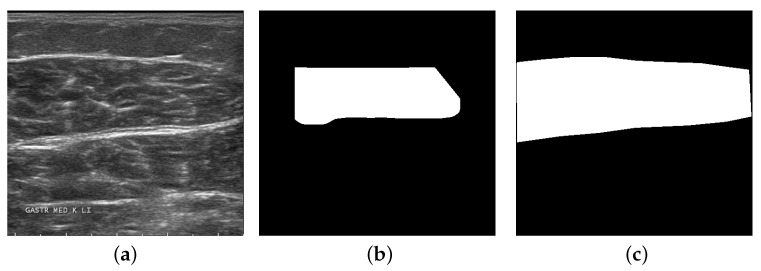
Comparison of the original segmentation label and the re-labeled label. The sample of ultrasound images (**a**), the original segmentation label (**b**), and relabeled segmentation label (**c**).

**Figure 7 entropy-25-00662-f007:**
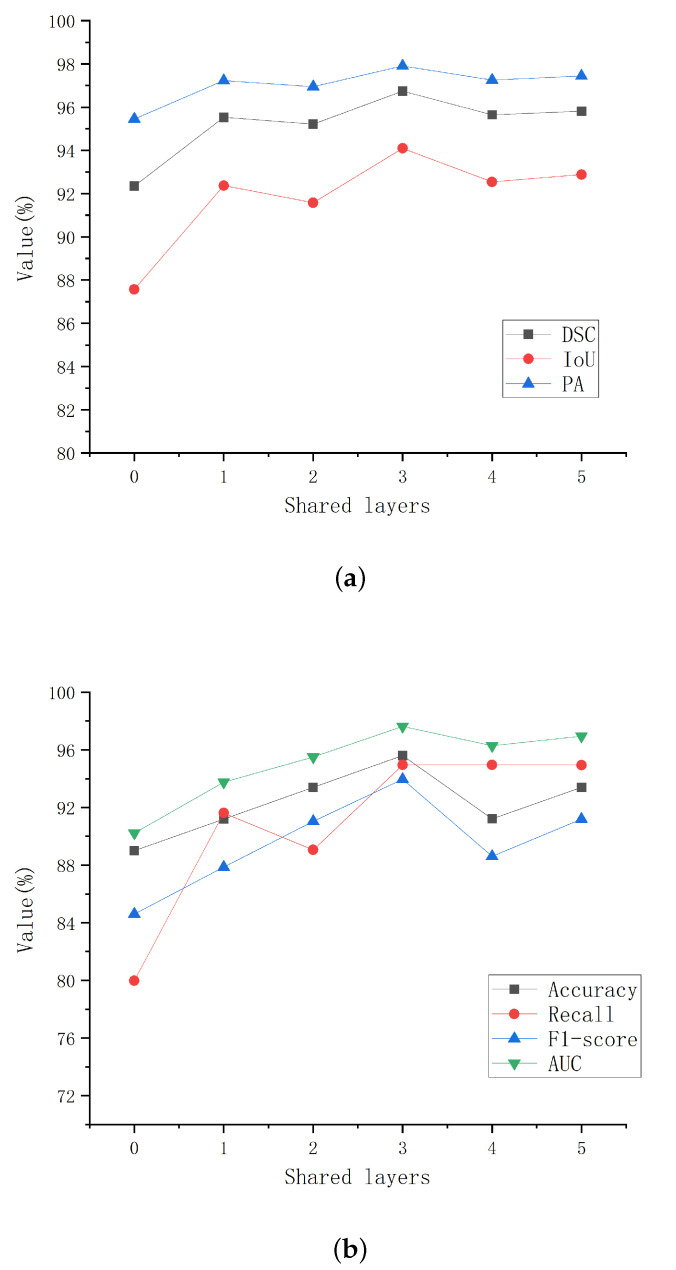
Comparison of model performance results for different shared layers. The results of segmentation (**a**) and the results of classification (**b**).

**Figure 8 entropy-25-00662-f008:**
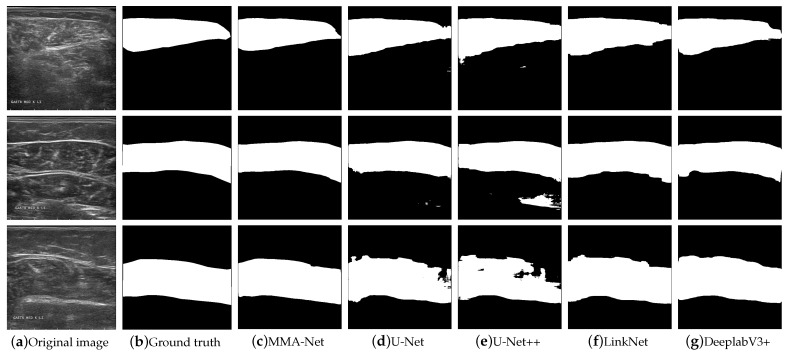
Segmentation results of healthy skeletal muscle transverse ultrasound images on different models.

**Figure 9 entropy-25-00662-f009:**
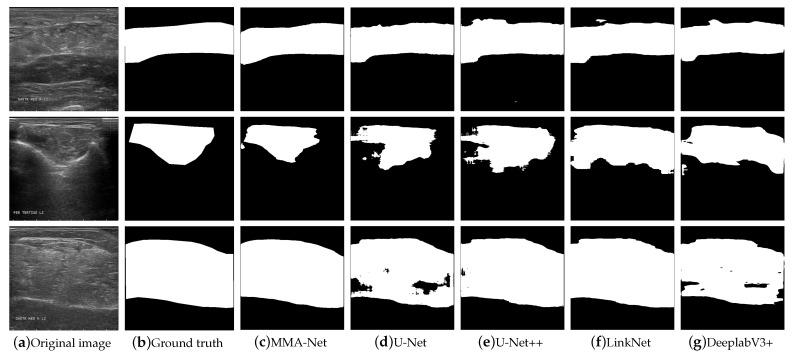
Segmentation results of diseased skeletal muscle transverse ultrasound images on different models.

**Figure 10 entropy-25-00662-f010:**
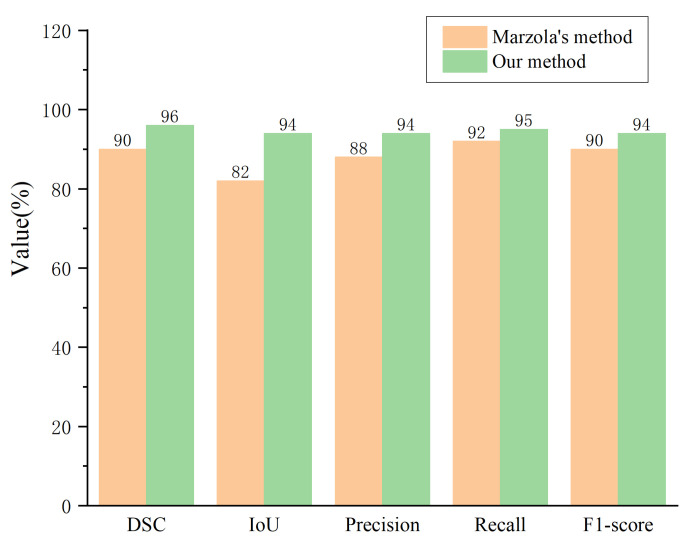
The bar plots of each evaluation index.

**Table 1 entropy-25-00662-t001:** Environment of experiments.

Name	Versions
CPU	Intel(R) Xeon(TM) Silver 4210R CPU@2.40GHz
GPU	NVIDIA GeForce RTX 3090
Operating System	Ubuntu 22.04 LTS
Framework	PyTorch 1.11.0
Language	Python 3.8.8

**Table 2 entropy-25-00662-t002:** Comparison of single-task and multi-task backbone models.

Task	Metrics	Single-Task	Multi-Task Backbone
Segmentation	DSC(%)	92.35	92.92
	IoU(%)	87.57	87.84
	PA(%)	95.45	95.61
Classification	Accuracy(%)	89.01	91.21
	Recall(%)	79.98	90.24
	F-Score(%)	84.61	87.91
	AUC(%)	90.22	94.19

**Table 3 entropy-25-00662-t003:** Segmentation results of ablation experiments for each module.

Models	DSC (%)	IoU (%)	PA (%)
Backbone	92.92	87.84	95.61
Backbone+ASPP	96.19	93.20	97.54
Backbone+ASPP+CA	96.13	93.34	97.60
Backbone+ASPP+CA+Connection(MMA-Net)	96.74	94.10	97.91

**Table 4 entropy-25-00662-t004:** Classification results of ablation experiments for each module.

Model	Accuracy (%)	Recall (%)	F-Score (%)	AUC (%)
Backbone	91.21	90.24	87.91	94.19
Backbone+ASPP	93.40	91.04	90.83	95.88
Backbone+ASPP+CA	95.60	91.04	93.77	96.57
Backbone+ASPP+CA+Connection (MMA-Net)	95.60	94.96	93.95	97.62

**Table 5 entropy-25-00662-t005:** The results of 10-fold cross validation on the MMA-Net.

Metrics	Mean (%)	SD (%)
**DSC**	96.57	1.17
**IoU**	94.84	2.03
**Accuracy**	96.29	1.75
**Recall**	95.21	1.97

**Table 6 entropy-25-00662-t006:** Comparison of other segmentation and classification models with the proposed model.

Models	Segmentation Results			Classification Results		
	DSC (%)	IoU (%)	PA (%)	Acc (%)	F-Score (%)	AUC (%)
U-Net	92.35	92.43	95.45	-	-	-
U-Net++	92.43	87.06	94.79	-	-	-
LinkNet	94.73	91.47	96.72	-	-	-
DeeplabV3+	94.67	90.92	96.46	-	-	-
VGG16	-	-	-	89.01	81.91	88.16
Resnet50	-	-	-	87.91	84.19	92.38
GoogleNet	-	-	-	92.30	87.91	93.71
MMA-Net	96.74	94.10	97.91	95.60	93.95	97.62

**Table 7 entropy-25-00662-t007:** Statistical significance analysis between MMA-Net and other segmentation models by *t*-test.

Model	Mean Difference ± Standard Error	95% CI	*p*-Value
U-Net	3.83 ± 0.49	[2.62,5.03]	*p* = 0.0002
U-Net++	4.01 ± 0.77	[2.78,5.83]	*p* < 0.0001
LinkNet	1.85 ± 0.43	[0.79,2.92]	*p* = 0.0053
DeeplabV3+	1.33 ± 0.32	[0.52,2.14]	*p* = 0.0068

**Table 8 entropy-25-00662-t008:** Statistical significance analysis between MMA-Net and other classification models by *t*-test.

Model	Mean Difference ± Standard Error	95% CI	*p*-Value
VGG16	5.06 ± 1.85	[0.53,9.59]	*p* = 0.0341
Resnet50	8.37 ± 1.12	[5.64,11.11]	*p* = 0.0003
GoogleNet	2.90 ± 0.97	[0.52,5.27]	*p* = 0.0243

**Table 9 entropy-25-00662-t009:** Comparison of the proposed method with existing analytical methods.

Analysis Content	Metrics	Our	Marzola
Segmentation of CSA	DSC	0.96	0.90
	IoU	0.94	0.82
Classification of abnormal muscles	Precision	0.94	0.88
	Recall	0.95	0.92
	F-score	0.94	0.90

## Data Availability

All the data and code used to support the findings of this study are available from the corresponding author upon request.

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
