# Peer review of "Automatic Analysis of Transverse Musculoskeletal Ultrasound Images Based on the Multi-Task Learning Model"

_entropy, 2023, doi:10.3390/e25040662_

Round 1

Reviewer 1 Report

Automatic analysis of transverse musculoskeletal ultrasound images using multi-task learning model

The research work is good, however there are some issues to be addressed:

1.       In the introduction has too much information. You should clearly state the following points:

a.       The problem of this research work.

b.       The contributions of this paper.

c.       The objectives of this work

d.       The guide to the rest of the other sections in this paper.

2.       Create another section, name it Literature work. Remove all the materials related to research work from the introduction section to the literature work section.

3.       Section 3 is Methods and so on.

4.       Improve figures as some of are very tiny.

5.       Equations have to be all cited if they are not yours.

6.       Are you results statistically significant, if yes, then you should prove it using some statistical methods.

Reviewer 2 Report

The authors present a new and promising method based on a multi-task deep-learning model to apply classification and segmentation to musculoskeletal ultrasound images. The article is in general well written and clear, and the results support the quality of the new method. Please find hereafter the remarks regarding the article.

Introduction:

- "Quantitative analysis is not only tedious and time-consuming, but also difficult to guarantee the accuracy of the results" <- Adding a citation here would be beneficial

- "then obtained quantitative or qualitative evaluation results through calculation" <- the meaning of "calculation" here is not very clear

- "It is validated by simulation experiments and real data to perform well for images with high scattered noise" <- from the sentence one could understand that the authors of [6] have chosen to apply the method on specific data to get good results (cherry picking) which is difficult to know, thus it could be better to reframe the sentence, e.g., "This method performed well on simulated and real images with high scattered noise" 

- "a heuristic algorithm is used" (line 63) <- More details are needed, which kind of heuristic algorithm?

- "intrinsic laws of data" <- what do you mean by that? Data-drive automatic feature extraction?

- "With the development of deep learning, single-task models are not sufficient to interpret complex medical images" <- the logical connection between the development of deep learning and the fact that single-task models are not sufficient should be developed

Experiments:

- " in the paper by Marzola et al." (line 267 and line 286 and 447) cite properly the article

- Reformulate the caption of Figure 6, so the letters appear after the  description of the image as in Figure 5.

- "Therefore, in this paper, based on this data, we re-labeled the images " -> Did experts do re-labeled the images? Can you provide more details regarding the re-labeling?

-  "All experiments are conducted on the server." <- make a reference to table 1 to check the details.

- More details regarding the choice of meta-parameters should be provided

- "3.3. Evaluation metrics" <- cite reference articles or text-books 

Results

- "4.1. Ablation Study" <- are the scores measured on the validation or test set? The authors state that "we finally constructed  the MMA-Net with a segmentation pixel accuracy of 97.91% and classification F-score and 366 AUC of 93.95% and 97.62%, respectively", if they update the structure of the network to improve the quality on the test set, then the test set is not longer valid, and it becomes a validation set. Please clearly define the scores on the validation, as well as the scores on the test sets. Why not using a cross-validation strategy, in order also to compare the distributions of scores by means of statistical tests? This point is capital and should clearly be addressed.

- Reformulate the caption of Figure 7, so the letters appear after the  description of the image as in Figure 5.

Discussion:

- Authors state that "The characteristics of ultrasound images are highly dependent on the ultrasound equipment and setup,  as well as the physical characteristics of the acoustic array, such as frequency and beam profile" -> How does your system is expected to behave to such kinds of perturbations in the data? 

English and proofreading, e.g., :

- "Musculoskeletal ultrasound images is an important basis" <- "Musculoskeletal ultrasound imaging is an important basis"

- "Experimentation with a dataset of medial transverse ultrasound images of the gastrocnemius muscle acquired from multiple subjects. " <- the sentence is incomplete

-  "extraction.It"<- "extraction. It" (line 53)

- "approach to learning and prediction" <- 'approach to learn and prediction'

- "is shown in Figure4." <- "is shown in Figure 4."

- ". he input image size" <- '. The input image size" line 298

- Some citations were not linked correctly:  "SGD[? ]" line 299

Reviewer 3 Report

The authors propose a multi-task deep learning method that is able to automatically segment transverse musculoskeletal ultrasound images and perform abnormality diagnose. However, it is unclear why the transverse ones matter, and what different information can be provided by the transverse and longitudinal musculoskeletal ultrasound images. The ultrasound images were from publicly available dataset but were relabeled. There was no mentioning of involvement of health professionals in this study. There was neither a single co-author with clinical background nor any mentioning of health professionals in the acknowledgement for the preparation or reassurance of new data labels, which however is important as human labels define the ground truths for the model evaluation. There was no guarantee that the new labels introduced by the authors were clinically relevant without participation of ultrasonographer/radiologists or physicians who were acquainted with musculoskeletal ultrasonography.

There was no description about the training and testing data in the abstract. How the division of the complete dataset was done? Was the split done by random selection or handpicking?

Methodologically, combining multi-scale feature fusion with an attention mechanism is not new. And the authors acknowledge that their novelty (line 138-141) is that they have applied multi-task learning (combining segmentation and classification, which is also common in many studies) to the analysis of transverse ultrasound images of skeletal muscles. It seems to me that the manuscript does not provide sufficient technological advancement but takes a not-uncommon combination of techniques for a specific clinical problem without clear explanation of why the clinical problem matters. In this regard, it should be more a clinical application-oriented manuscript than a technique -oriented one. But it is written in a form of the latter.

Concerning the results, there was no single statistical test to check whether any numerical differences are statistically significant or not. The authors showed only the average values without providing any information of their distributions. Besides, what are presented in Figure 10 are just bar plots, not histograms. There was also no discussion of potential risks associated with their model, for instance, over-fitting.

Round 2

Reviewer 1 Report

The authors have done  a significant revise.

Author Response

We are very grateful to your comments. We have made further revisions to the manuscript, and hope that the modifications will meet with approval.

Reviewer 2 Report

The authors has successfully addressed the comments in details.

Author Response

(The authors gave the same response as above.)

Reviewer 3 Report

I am Chinese-literate but have not found a hospital with a name that can be matched to “Qingdao University Applied Hospital” or “Applied Hospital of Qingdao University” in English. It is seems to me that the authors have made up the organization and the health professional whom they have now put in acknowledgement in order to convince readers that the data-relabeling was performed by capable health professionals. This is a very serious issue, and it makes the entire study not trust-worthy to me. Besides, there was no mention of what labeling software was used, what the experience level, i.e., how many years of experiences in reading musculoskeletal ultrasound images Doctor Lili Xu has.

Performing box-plots should not be considered equivalent to a statistical test that can reject the null hypothesis with sufficient confidence that the observed differences in two randomly sampled groups is not by chance. Given the box-plots only for segmentation evaluation in Figure 8, no statistical significance can be guaranteed. Judging from my personal experience, simple student t-tests on the data will very likely show that the p-values will most likely be >0.05 at least for the dice coefficient.

The musculoskeletal disease state classification is more important. In my view, segmentation task is just auxiliary to the classification task. Even if the model performs better in segmentation (this is not proven yet as no statistical tests were performed), but if the p-values for group comparison show that the final classification by the MMA-Net is not statistically better, the improvement is not very much meaningful.

Round 3

Reviewer 3 Report

I  have tested translation services provided by Bing and Baidu which are both assessable in China while none of them make a mistake between "affiliated'' and "applied". Starting with the same letter "a" does not make their meanings to be anywhere close. If the authors know what they are writing and properly proofread their writings, this kind of mistake can never happen. It does not require a linguist to identify such an obvious mistake. However , I have neither adequate knowledge in the musculoskeletal system nor expertise in interpreting the ultrasound images of it to judge whether the authors have made similar mistakes in their writings for the medical science part or not. I don't know either whether this manuscript has been reviewed by medical professionals specialized in this area. I guess not as entropy is not a medical journal, making it hard for the editorials to find reviewers with the specialized medical expertise in this area. And it is very problematic that none of the authors have documented medical background in this particular field to hold accountable for the correctness of the content, except Doctor Lili Xu. However as she is not listed as an author to hold accountability for the medical content, we may face the worst scenario that no one can identify  obvious errors in the medical content similar to mistaking "applied" for "affiliated" that the authors might have made.

I should also stress that it is erroneous for the authors to write "Therefore, in this paper, based on this data, we re-labeled the images used FIJI software[ 38] according to the definition of muscle cross-sectional area". The listed authors should not state "we". They did not do it, and they were not qualified to do it. If they did it, the whole manuscript should not be trusted and must not be accepted for publication. I am sorry about taking a strong opinion on this, but expertise in each specialized area must be respected.

Besides, it is not recommended to use the average and standard deviation in table 7 and 8 for the statistical evaluation without verifying whether the statistical distribution of a parameter follows a normal distribution first.  If not, use the median and quartiles instead. 
